# Label-Free Fluorescent Turn-On Glyphosate Sensing Based on DNA-Templated Silver Nanoclusters

**DOI:** 10.3390/bios12100832

**Published:** 2022-10-06

**Authors:** Yuliang Cheng, Guowen Li, Xiufang Huang, Zhijuan Qian, Chifang Peng

**Affiliations:** 1State Key Laboratory of Food Science and Technology, Jiangnan University, Lihu Road 1800, Wuxi 214122, China; 2School of Food Science and Technology, Jiangnan University, Lihu Road 1800, Wuxi 214122, China; 3Nanjing Customs District Light Industry 375 Productsand Children’s Products Inspection Center, Yangzhou 225009, China; 4International Joint Laboratory on Food Safety, Jiangnan University, Lihu Road 1800, Wuxi 214122, China

**Keywords:** label free, fluorescence, turn on, DNA, Cu^2+^, silver nanoclusters

## Abstract

In this work, a label-free fluorescent detection method for glyphosate, based on DNA-templated silver nanoclusters (DNA-Ag NCs) and a Cu^2+^-ion-modulated strategy, was developed. In the presence of Cu^2+^, the fluorescence of the DNA-Ag NCs was quenched. Glyphosate can restore the fluorescence of DNA-Ag NCs. By analyzing the storage stability of the obtained DNA-Ag NCs using different DNA templates, specific DNA-Ag NCs were selected for the construction of the glyphosate sensor. The ultrasensitive detection of glyphosate was achieved by optimizing the buffer pH and Cu^2+^ concentration. The sensing of glyphosate demonstrated a linear response in the range of 1.0–50 ng/mL. The limit of detection (LOD) was 0.2 ng/mL. The proposed method was successfully applied in the detection of glyphosate in a real sample, indicating its high application potential for glyphosate detection.

## 1. Introduction

Glyphosate, a broad-spectrum organophosphorus herbicide, is the largest pesticide by sales in the global crop protection market. It is widely used around the world and mainly used for weed removal before crop planting. There is increasing evidence that glyphosate is potentially toxic to non-target organisms [1]. In 2017, the International Agency for Research on Cancer (IARC) classified glyphosate as a possible human carcinogen [2]. Glyphosate applied in agricultural environments can enter the water environment through various ways [3]. In recent years, glyphosate has been found in surface water worldwide. Therefore, the determination of glyphosate in water is very important.

Some conventional analytical methods such as high-performance liquid chromatography [4] and mass spectrometry [5] are commonly used for the determination of pesticide residues in foods. Although these methods provide accurate and sensitive results, we have to tolerate their disadvantages, including their complicated operation and requirement of professional personnel. Therefore, there is a growing need to establish simple, rapid, sensitive, and low-cost sensing methods for glyphosate, which benefit on-site detection in resource-limited scenarios.

DNA template nanosensors exhibit excellent optical properties, including high-fluorescence quantum yield and stability, biocompatibility, ease of synthesis, and low toxicity. By changing the DNA sequence and structure, the DNA template nanosensors could be used for the detection of different targets [6]. The various DNA template silver nanoclusters (DNA-Ag NCs) with fluorescence emission from the UV to near-infrared regions could also be synthesized based on changing the DNA sequence and environmental factors [7]. DNA-Ag NCs have been applied in the detection of a variety of targets, such as heavy metal ions [8], proteins [9], viruses, microRNAs [10], and thiols [11]. However, few studies have been involved in the application of DNA-Ag NCs in the enzyme-free fluorescent detection of pesticides.

Herein, we developed a fluorescent detection glyphosate based on DNA-Ag NCs. In the presence of Cu^2+^, which can bind to the DNA template through the electrostatic interaction with the phosphate group and basic bases, the fluorescence of the DNA-Ag NCs was quenched. However, glyphosate can trap Cu^2+^ and greatly restore the fluorescence of DNA-Ag NCs. Thus, the DNA-Ag NC-based “turn-on” fluorescent sensing of glyphosate was realized (Figure 1). Moreover, we found specific DNA-templated Ag NCs that demonstrated excellent storage stability and were suitable for the above glyphosate sensing strategy.

## 2. Materials and Apparatus

### 2.1. Chemicals and Reagents

All chemicals were used directly without any further purification, and all chemical reagents in this experiment were of analytical grade. Sodium borohydride (NaBH_4_), silver nitrate (AgNO_3_), zinc nitrate hexahydrate (Zn(NO_3_)_2_·6H_2_O), iron nitrate hexahydrate (Fe(NO_3_)_3_·9H_2_O), lead nitrate (Pb(NO_3_)_2_), nickel sulfate hexahydrate (NiSO_4_·6H_2_O), manganese chloride tetrahydrate (MnCl_2_·4H_2_O), aluminum nitrate hexahydrate (Al(NO_3_)_3_·9H_2_O), mercury nitrate (Hg(NO_3_)_2_), cobalt nitrate (Co(NO_3_)_2_), ethylene diamine tetra-acetic acid (EDTA), 5-morinopropanulfonic acid (MOPs), and sodium hydroxide (NaOH) were purchased from Shanghai Aladdin Biochemical Technology Co., Ltd. (Shanghai, China). All experimental water resistance values were higher than 18 MΩ/cm. All pesticides were purchased from Shanghai Pesticide Research Institute (Shanghai, China). The template DNA was synthesized by Sangon Bioengineering (Shanghai, China) Co. Sangon Bioengineering Co., Ltd. (Shanghai, China)

### 2.2. Apparatus

The transmission electron microscope (TEM) images of DNA-Ag NCs were obtained using a JEOL-2100 transmission electron microscope (Japan Electron Optics Laboratory Co., Ltd., Tokyo, Japan). The fluorescence spectra were measured using a F97Pro fluorescence spectrophotometer (Shanghai Ling Guang Technology Co., Ltd., Shanghai, China). The circular dichroism (CD) spectra were recorded by Chirascan V100 (Applied Photophysics, Surrey, UK). The weight measurements were carried out via analytical balance (MS105DU, Mettler Toledo Instruments Shanghai Co., Ltd., Shanghai, China).

## 3. Experimental Method

### 3.1. Preparation of DNA-Ag NCs

The DNA-Ag NCs were synthesized through the one-step method. Briefly, 16 µL of DNA (250 µmol /L) was mixed with 166 µL of phosphate buffer (PB, 20 mM, pH = 7.0) and 6 µL of AgNO_3_ (4 mM) under vigorous stirring, and then the mixture was incubated at 4 °C for 20 min. Then, 12 µL of NaBH_4_ (2 mmol/L) prepared with ice water was added to the mixture under vigorous stirring and then incubated at room temperature for 3 h in the dark. Finally, the obtained DNA-Ag NCs were stored at 4 °C for future use.

### 3.2. Protocol of the Glyphosate Detection

Different concentrations of glyphosate (5 µL) were mixed with 20 µL of Cu^2+^ (300 nM) in 10 mM MOPS buffer (pH 7.5) and incubated for 2 min. Subsequently, 60 µL of DNA-Ag NC solution and 15 µL of MOPS buffer (pH 7.5) were added. The final volume of the mixture was 100 µL. After incubation for 25 min at the room temperature, the fluorescence intensity of the mixture was recorded using a fluorescence spectrophotometer.

### 3.3. Glyphosate Detection in Real Samples

Tap water and spring water were collected from the lab and supermarket as real samples. Different concentrations of glyphosate (5 ng/mL, 20 ng/mL, and 40 ng/mL) were added to the samples and filtered with a 0.22 μm microporous membrane. The fluorescence emission spectra were recorded and the recovery rates were calculated.

## 4. Results and Discussion

### 4.1. Sensing Strategies for Glyphosate Detection

The interactions of metal cations with nucleic acid have been used in the design of DNA-based nanosensors [12]. It was reported that Cu^2+^ was mainly attached to the phosphate group of nucleic acids and also can bind to the basic groups of nucleic acid [13]. Cu^2+^ mainly binds to the N7 and O6 positions of guanine and the O2 position of cytosine [14]. Ag^+^ can specifically bind to the N3 site of cytosine [15]. Compared with adenine (A), guanine (G), and thymine (T) bases, Ag^+^ towards the cytosine (C) base showed much a higher binding affinity. The binding constants of C-Ag^+^-C base pairs can be compared with T-Hg^2+^-T base pairs [16].

Based on the binding of Cu^2+^ to nucleic acid, we designed DNA-Ag NCs as a fluorescent probe to achieve rapid glyphosate detection through Cu^2+^-mediated fluorescence modulation.

Cu^2+^ can bind to phosphate and basic groups in DNA, quenching the fluorescence of nanoclusters. In the presence of glyphosate, due to the strong binding affinity of the phosphonyl group (-PO_3_H_2_) and carboxyl group (-COOH) with Cu^2+^, Cu^2+^ was trapped and the fluorescence of the DNA-Ag NCs was greatly restored.

### 4.2. Characterization of DNA-Ag NCs

The sequence and structure of the DNA template could greatly affect the fluorescence characteristic of DNA-Ag NCs [17]. For example, Dickson et al. obtained Ag NCs using five kinds of single-stranded DNA with similar sequences [18]. Yang et al. obtained silver nanoclusters with different emission wavelengths by adjusting the structural changes of the DNA template between a stem–loop structure and dimer [19]. The C-rich DNA sequence 5′-CCCTTAATCCCC-3′ was usually used as a template for the synthesis of DNA-Ag NCs. It was found that the G base could enhance the fluorescence of the C-rich DNA-Ag NCs. Therefore, we selected five typical C-rich DNA sequences reported in the literature. Two of them were inserted with multiple G bases (Table 1). They were evaluated for subsequent experiments.

As shown in Table 2, the maximum excitation wavelength and the maximum emission wavelength of the five DNA-Ag NCs showed great differences. The fluorescence intensity of the DNA2-Ag NCs, DNA3-Ag NCs, and DNA4-Ag NCs was higher than the other two Ag NCs (Figure 2), which was clearly observed under the 365 nm UV lamp (Figure 3).

The stability of Ag NCs is important for their practical applications. As shown in Figure 4, three kinds of DNA-Ag NCs were compared in terms of their stability. It was found that the fluorescence intensity of the DNA2-Ag NCs remained stable after being stored for 16 days. However, the fluorescence intensity of the DNA3-Ag NCs and DNA4-Ag NCs decreased by about 50% after 5 days. Therefore, we selected the DNA2-Ag NCs for the subsequent glyphosate detection.

The morphology and particle size of the DNA2-Ag NCs were characterized via TEM and DLS. As shown in Figure 5, the synthesized DNA2-Ag NCs had no aggregation and the average particle size was about 2.2 nm.

As shown in Figure 6, the maximum excitation peak (λ_ex_) of the DNA2-Ag NCs was at 530 nm and the maximum emission peak was at 620 nm.

### 4.3. Feasibility Verification of Glyphosate Detection

As shown in Figure 7, the DNA2-Ag NCs demonstrated a red fluorescence emission under 530 nm excitation. With the addition of the Cu^2+^ solution (60 nM), the fluorescence of the DNA2-Ag NCs was sharply quenched. In the presence of the glyphosate solution (500 ng/mL), the fluorescence of the DNA2-Ag NCs was significantly restored. However, the glyphosate alone had no significant effect on the fluorescence emission of the Ag NCs. Therefore, it is feasible to use Cu^2+^ to mediate the fluorescent detection of glyphosate based on the Ag NCs.

### 4.4. Optimization of Sensing Conditions

The Cu^2+^ concentration towards the fluorescence quenching of the DNA2-Ag NCs was studied. As shown in Figure 8A,B, when the Cu^2+^ concentration was between 5 nM and 140 nM, the DNA2-Ag NCs quenching (F/F_0_) gradually increased with the increased Cu^2+^ concentration (F_0_ and F refer to the fluorescence intensity of the DNA2-Ag NCs in the absence and presence of Cu^2+^ ions, respectively). Meanwhile, the quenching efficiency ((F_0_ − F)/F_0_) reached 90% when the Cu^2+^ concentration was higher than 60 nM, which indicated that the fluorescence of the DNA2-Ag NCs was sensitive to the change in Cu^2+^ concentration. To further investigate the mechanism of the quenching effect, the Stern–Volmer equation was used [20]:(1)$$\frac{F_{0}}{F}=1+K_{SV}[Q]=1+k_{q}\tau_{0}$$

Here, F_0_ and F are the fluorescence intensity in the absence and presence of the quencher (Cu^2+^), respectively; K_sv_ is the Stern–Volmer quenching constant, [Q] is the quencher concentration, and k_q_ is the quenching rate constant of the DNA-Ag NCs; τ_0_ is the average excited-state lifetime of the DNA-Ag NCs, reported as 2.23 ns [18]. The linear regression of the F_0_/F plot versus [Q] determines the K_sv_ value. As shown in Figure 8C, the Stern–Volmer plot is linear and the value of K_sv_ is 0.139. The k_q_ was calculated using the following equation:(2)$$k_{q}=\frac{K_{sv}}{\tau_{0}}$$
and the value of k_q_ was calculated to be 6.2 × 10^16^ M^−1^ s^−1^, which was much higher than the maximum dynamic quenching constant (2.0 × 10^10^ M^−1^ s^−1^)[21]. Thus, the dynamic quenching was not predominated, since a complex was formed between the Cu^2+^ and DNA-Ag NCs. Therefore, static quenching was the predominant mechanism of the DNA-Ag NCs quenching in the presence of Cu^2+^, which resulted in the complex formation.

In addition, the binding parameters of Cu^2+^ to DNA-Ag NCs were calculated using the following equation [22]:(3)$$\log[\frac{F_{0}-F}{F}]=\log K+n\log[Q]$$
where n is the number of binding sites and K is the binding constant. As shown in Figure 8D, based on the plot of the double log graph of [(F_0_ − F)/F] versus log [Q], the values of n and K were obtained from the slope and Y-intercept, respectively. The result indicated that the binding constant between the DNA-Ag NCs and Cu^2+^ was 3.2 × 10^7^ M^−1^, which indicated that Cu^2+^ could bind to DNA-Ag NCs effectively. Madsen et al. analyzed the stability constants of several 1:1 metal complexes of glyphosate [23], and the stability constant (LogK_ML_) for the Cu^2+^-glyphosate complex was 11.92. Thus, the glyphosate binding to Cu^2+^ is much higher than for DNA-Ag NCs binding to Cu^2+^. The stronger binding of the glyphosate to Cu^2+^ provided feasibility for the glyphosate sensing.

To evaluate the influence of other metal ions on the glyphosate detection, Zn^2+^, Fe^3+^, Pb^2+^, Ni^2+^, Mg^2+^, Mn^2+^, Ca^2+^, Al^3+^, Hg^2+^, and Co^2+^ at a ten times higher concentration (600 nM) than Cu^2+^(60 nM) were tested under the same conditions. As shown in Figure 9, the fluorescence of the DNa2-Ag NCs was greatly quenched by the Cu^2+^ and Hg^2+^. The results showed that the glyphosate could selectively bind to the Cu^2+^, and most metal ions would not interfere with the detection of glyphosate, except Hg^2+^. Since the pollution limit of Hg^2+^ in water is much lower than other metal ions, there is a low possibility of interference from Hg^2+^.

The effects of the pH and Cu^2+^ concentration on the sensitivity of the sensing system were investigated. As shown in Figure 10A, when the pH value increased from 6 to 7.5, the fluorescence recovery (F_2_ − F_1_)/F_2_ (F_1_ and F_2_ refer to the fluorescence of the DNA-Ag NCs/Cu^2+^ system in the absence and presence of glyphosate, respectively) gradually increased, and the highest fluorescence recovery rates were obtained at pH 7.5. The results indicated that the sensing system was more stable in weak alkaline environments. Therefore, Mops buffer (pH, 7.5) was selected for the subsequent experiments. The effect of the Cu^2+^ concentration was also investigated. As shown in Figure 10B, when the Cu^2+^ concentration was in the range of 40–100 nM, high fluorescence recovery of DNA-Ag NCs/Cu^2+^ could be obtained, and the highest recovery was obtained at 60 nM. Therefore, 60 nM Cu^2+^ was selected for glyphosate detection.

### 4.5. Analytical Performance of the Glyphosate Detection

As shown in Figure 11, the fluorescence recovery, (F_2_ − F_1_)/F_1_, gradually increased with the increased concentration of glyphosate in the range of 1.0 ng/mL~400 ng/mL, and a linear equation, y = 0.058x + 0.081 (R^2^ = 0.992), was obtained in the range of 1–70 ng/mL. The LOD was calculated to be 0.2 ng/mL based on 3σ/s (σ is the standard deviation of the blank value and s is the slope of the equation).

Compared with some other typical fluorescence methods for the detection of glyphosate in recent years (Table 3), our proposed fluorescence detection method of glyphosate demonstrates much better sensitivity. Although Huang et al. constructed an ultrasensitive glyphosate fluorescence detection method using papain-coated gold nanoclusters as fluorescence probes combined with a tyrosinase/dopamine system [24], this method was relatively complex due to introducing a tyrosinase amplification system.

### 4.6. Selectivity Analysis

To evaluate the selectivity of the proposed method, eight more common pesticides including isocarbophos, phosalone, dimethoate, chlorpyrifos, fenamiphos, imidacloprid, acetamidine, and carbofuran at a five times higher concentration (1250 ng/mL) than glyphosate (250 ng/mL) were tested under the optimal conditions. As shown in Figure 12, the DNA2-Ag NCs/Cu^2+^ fluorescence sensing system did not respond to these high concentrations of pesticides, which indicated the excellent selectivity of that sensing system.

### 4.7. Mechanism of Glyphosate Sensing

The quenching fluorescence of DNA-Ag NCs by Cu^2+^ was mainly ascribed to the interaction of Cu^2+^ with the phosphate base in the DNA [34]. To confirm this, EDTA, a strong Cu^2+^ chelator [35], was used to challenge the fluorescence quenching by Cu^2+^. As shown in Figure 13, the addition of EDTA restored most of the fluorescence emissions, which indicated that complexation was the main reason. The surfaces of Cu^2+^ and DNA2-Ag NCs neutralized part of the negative charge of the DNA template, thereby quenching the fluorescence of the DNA2-Ag NCs through electron or energy transfer processes [36]. At the same time, there may be another effect in the above fluorescence quenching, which we speculated was a metalphilic interaction between the copper and silver [37].

From the circular dichroism spectrum in Figure 14, the DNA2-Ag NCs have positive and negative absorption peaks at 275 nm and 245 nm, respectively, showing a typical B-type DNA conformation [14]. After the addition of Cu^2+^, the absorption peaks at 245 nm and 275 nm obviously redshifted, which was associated with the changes in the DNA template microenvironment [38]. This result confirmed the occurrence of the interaction between the Cu^2+^ and oligonucleotide chains. Therefore, we hypothesized that both the carboxyl group (-COOH) and phosphonacyl group (-PO_3_H_2)_ in the glyphosate chelated with Cu^2+^, which destroyed the interaction between Cu^2+^ and DNA2-Ag NCs, resulting in the recovery of the DNA2-Ag NCs’ fluorescence.

## 5. Conclusions

In conclusion, a fluorescence turn-on sensor for glyphosate detection was constructed using DNA-Ag NCs. It was based on the Cu^2+^-mediated strategy, in which Cu^2+^ can effectively quench the fluorescence of DNA-Ag NCs, and the coordination between the glyphosate and Cu^2+^ restored the fluorescence of the DNA-Ag NCs. The established method has the advantages of high sensitivity, simple operation, and low costs. The method was also used to detect glyphosate in tap water and spring water samples with satisfactory recovery. Our work provided a new option for the detection of glyphosate, and also showed that DNA-Ag NCs have high potential in pesticide detection.

## Figures and Tables

**Figure 1 biosensors-12-00832-f001:**
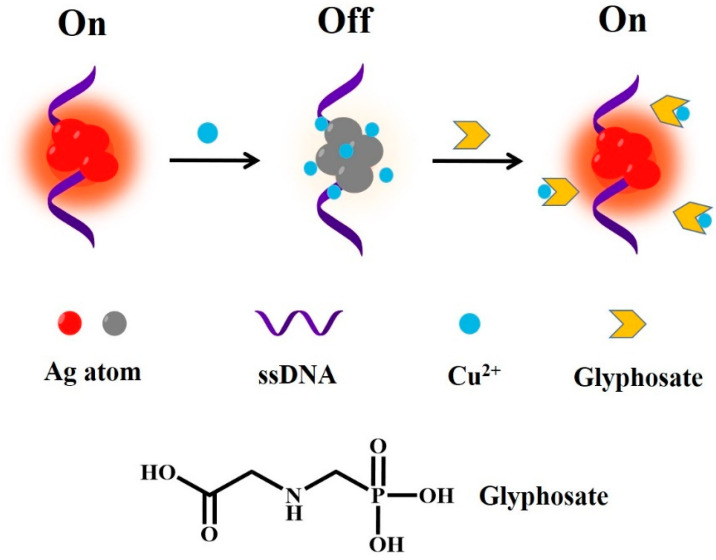
Schematic illustration of the detection of glyphosate based on DNA-Ag NCs.

**Figure 2 biosensors-12-00832-f002:**
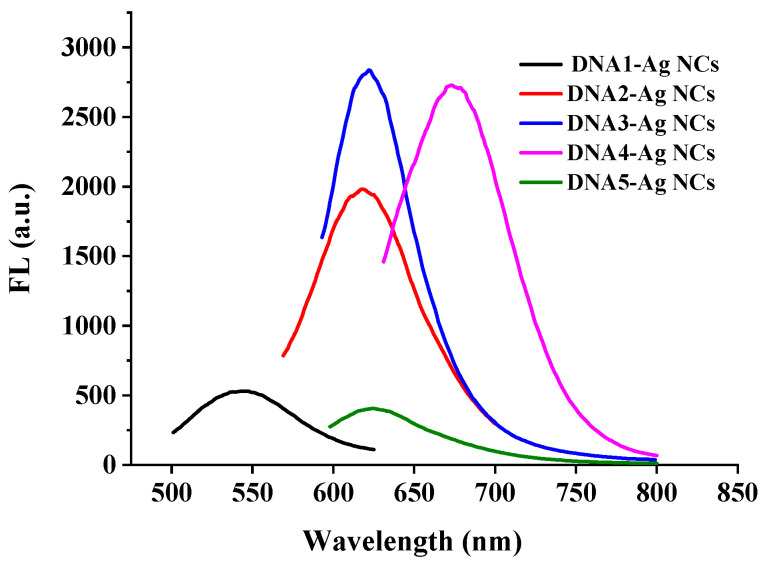
Fluorescence spectra of DNA-Ag NCs.

**Figure 3 biosensors-12-00832-f003:**
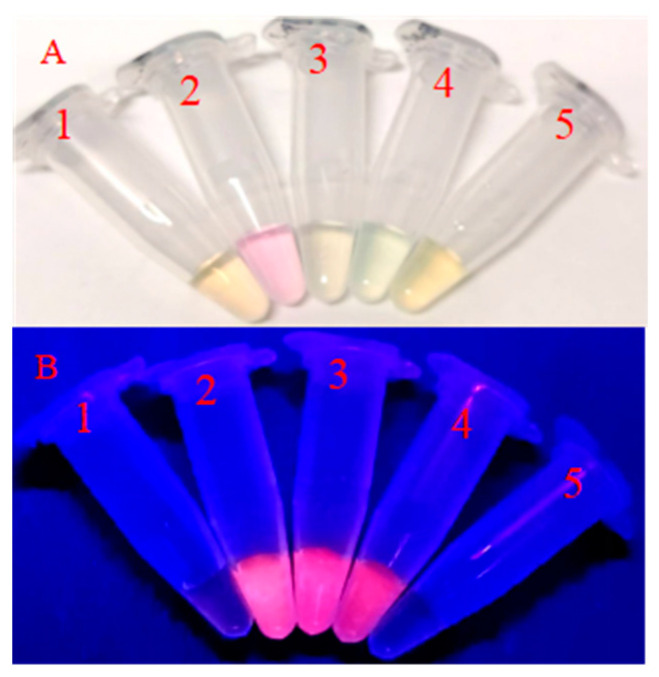
Digital images of DNA-Ag NCs illuminated with (**A**) white and (**B**) UV lights. 1–5 refer to the DNA1-Ag NCs to DNA5-Ag NCs.

**Figure 4 biosensors-12-00832-f004:**
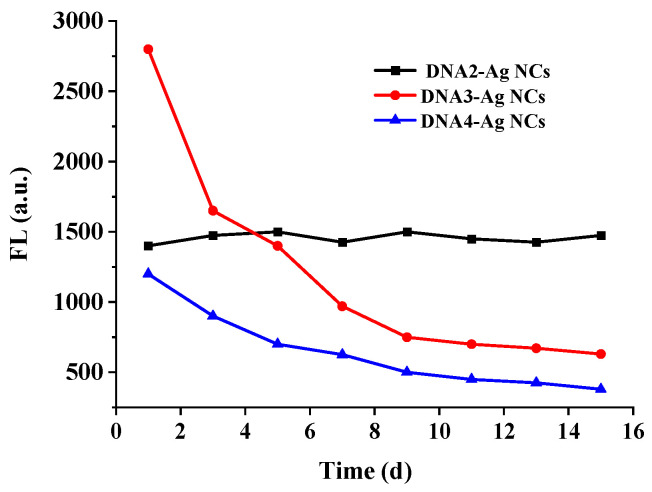
The stability of the prepared DNA-Ag NCs.

**Figure 5 biosensors-12-00832-f005:**
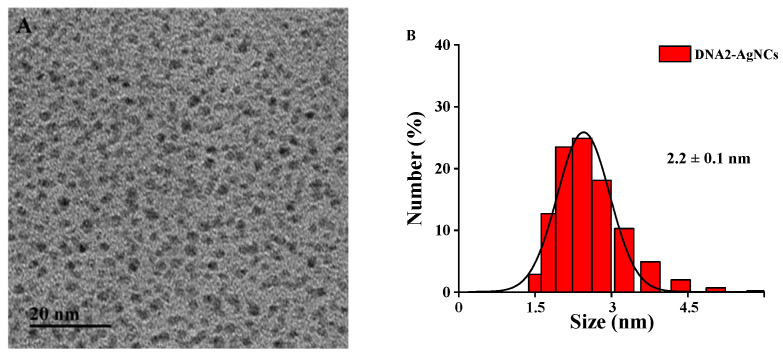
(**A**) TEM image of the DNA2-Ag NCs. (**B**) Particle size distribution histogram of the DNA2-Ag NCs.

**Figure 6 biosensors-12-00832-f006:**
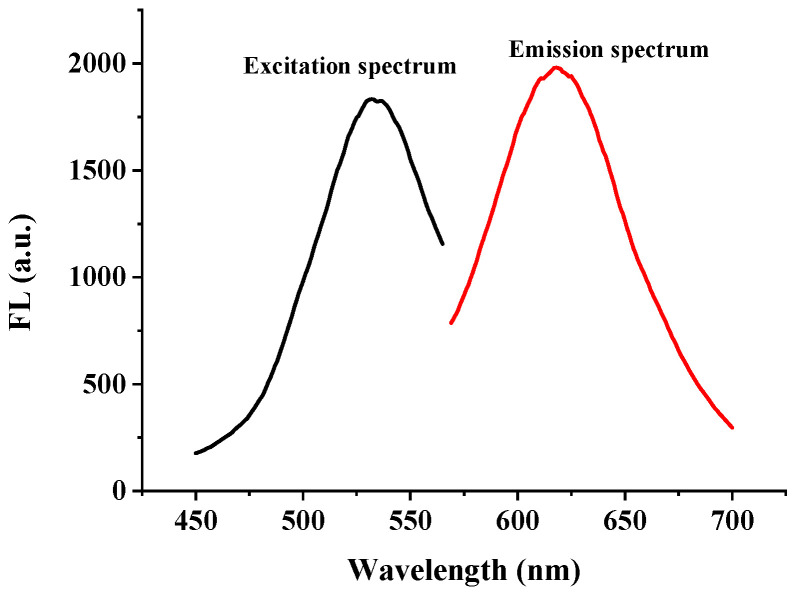
Fluorescence spectra of DNA2-Ag NCs.

**Figure 7 biosensors-12-00832-f007:**
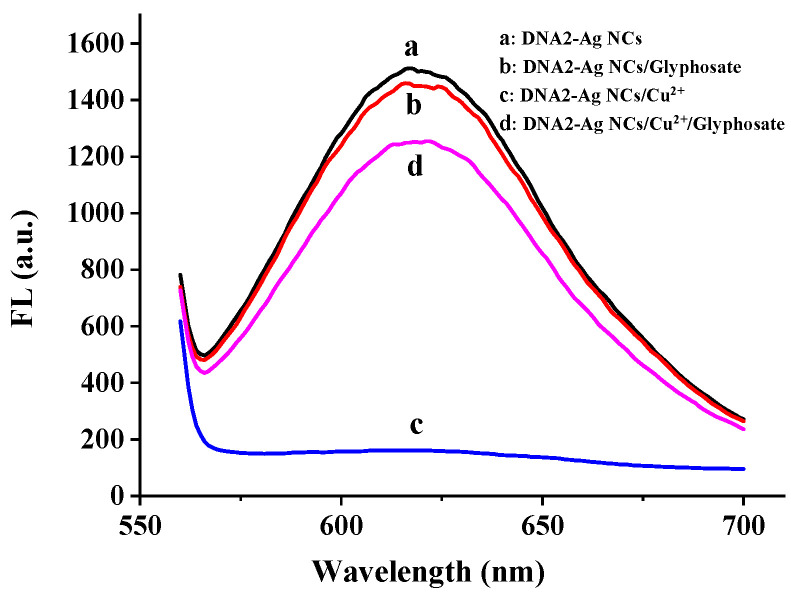
Fluorescence spectrum of DNA2-Ag NCs (a), DNA2-Ag NCs/glyphosate (b), DNA2-Ag NCs/Cu^2+^ (c), and DNA2-Ag NCs/Cu^2+^/glyphosate (d).

**Figure 8 biosensors-12-00832-f008:**
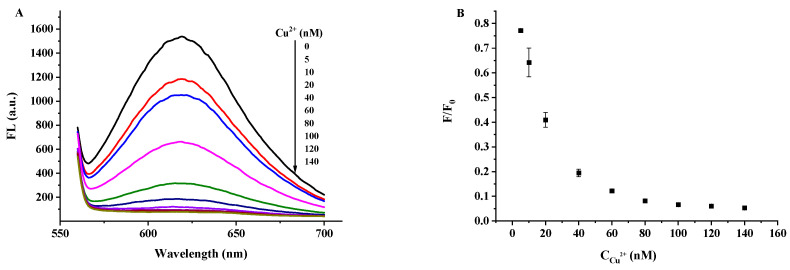
(**A**) The fluorescence quenching effect of DNA2-Ag NCs after adding Cu^2+^ (0 nM~140 nM). (**B**) The relationships between the F/F_0_ and different concentrations of Cu^2+^. (**C**) The Stern–Volmer plots of DNA2-AgNCs with Cu^2+^ F_0_ and F are the fluorescence intensities of the DNA2-Ag NCs in the absence and presence of Cu^2+^. (**D**) Computation of the binding constant (K) of each binding site and the number of binding sites of DNA-AgNCs (n).

**Figure 9 biosensors-12-00832-f009:**
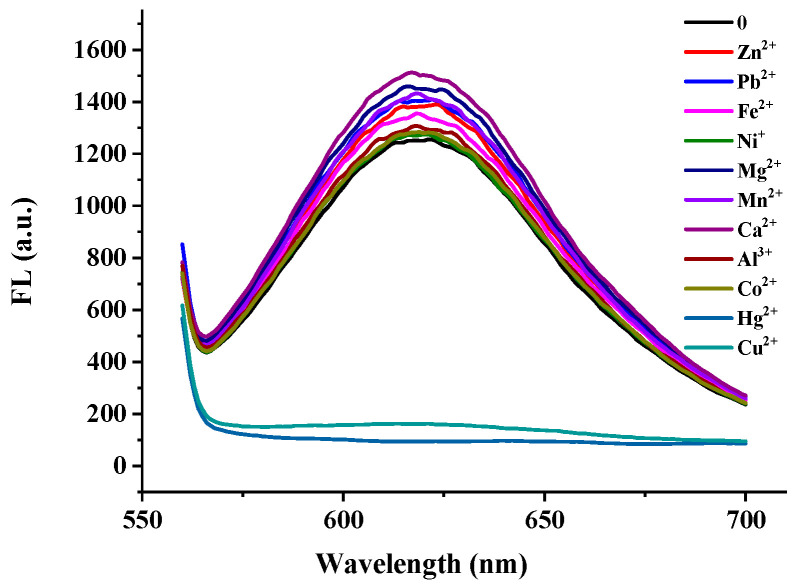
Fluorescence spectra of DNA2-Ag NCs in the presence of different metal ions.

**Figure 10 biosensors-12-00832-f010:**
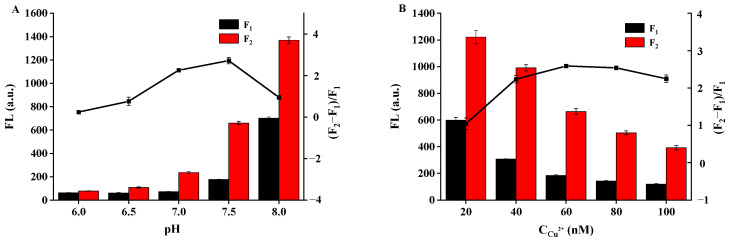
Effects of (**A**) pH and (**B**) different concentrations of Cu^2+^ on the recovery efficiency of DNA2-Ag NCs/Cu^2+^/glyphosate.

**Figure 11 biosensors-12-00832-f011:**
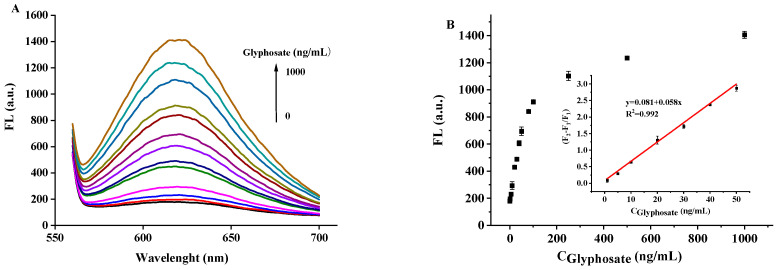
(**A**) The fluorescence recovery effect of DNA2-Ag NCs after adding different concentrations of glyphosate. (**B**) The relationship between the fluorescence intensity and different concentrations of glyphosate.

**Figure 12 biosensors-12-00832-f012:**
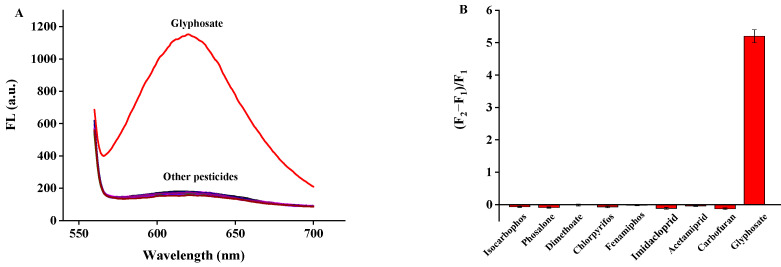
Selectivity of the assay for various pesticides. (**A**) Fluorescence; (**B**) pesticides.

**Figure 13 biosensors-12-00832-f013:**
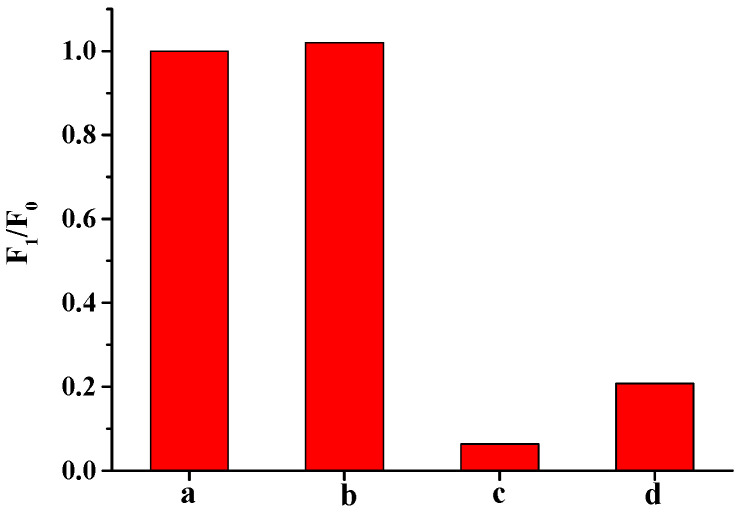
The fluorescence ratios of (a) DNA2-Ag NCs, (b) DNA2-Ag NCs/EDTA, (c) DNA2-Ag NCs/Cu^2+^, and (d) DNA2-Ag NCs/Cu^2+^/EDTA.

**Figure 14 biosensors-12-00832-f014:**
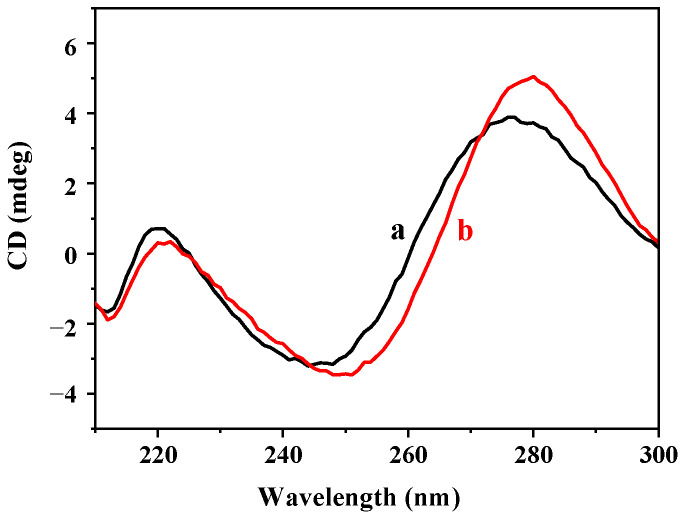
The circular dichroism spectra of (a) DNA2-Ag NCs and (b) DNA2-Ag NCs/Cu^2+^.

**Table 1 biosensors-12-00832-t001:** Sequences of five single-stranded DNA.

DNA	Sequence 5′-3′
DNA 1	*CCCTTAATCCCC*
DNA 2	ACCCGAACCTGGGCTACCA *CCCTTAATCCCC*
DNA 3	ATCCTCCCACC**GGG**CCTCCCACCATAAAAA *CCCTTAATCCCC*
DNA 4	GGCAGGTT**GGGG**TGACTAAAAA *CCCTTAATCCCC*
DNA 5	CTGACACCATATTATGAAGA *CCCTTAATCCCC*

**Table 2 biosensors-12-00832-t002:** Excitation and emission wavelengths of different DNA-Ag NCs.

Name	λ_ex_/nm	λ_em_/nm
DNA1-Ag NCs	464	550
DNA2-Ag NCs	530	620
DNA3-Ag NCs	560	621
DNA4-Ag NCs	596	671
DNA5-Ag NCs	560	627

**Table 3 biosensors-12-00832-t003:** Comparison of different methods used for detecting glyphosate.

Method	Linearity Range	LOD	Ref.
Electrochemistry	0.028~28 μg/mL	10 ng/mL	[25]
Chemiluminiscence	0.015~12 μg/mL	15 ng/mL	[26]
SERS	0.016~16 μg/mL	2.4 ng/mL	[27]
Fluorescence colorimetric	/	0.69 ng/mL	[28]
LFA	0.005~50 μg/mL	2 ng/mL	[29]
Fluorescence	0.1~1 μg/mL	7.8 ng/mL	[30]
Fluorescence	0.3~3 μg/mL	100 ng/mL	[31]
Fluorescence	0.04~0.4 ng/mL	0.035 ng/mL	[32]
Fluorescence	/	13 ng/mL	[33]
Fluorescence	1~50 ng/mL	0.2 ng/mL	This method

## Data Availability

All data generated or analyzed during this study are included in this published article.

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
