# Peer review of "Label-Free Fluorescent Turn-On Glyphosate Sensing Based on DNA-Templated Silver Nanoclusters"

_biosensors, 2022, doi:10.3390/bios12100832_

Round 1

Reviewer 1 Report

1 More characterization toward DNA2-Ag NCs should be given and discussed.

2 The label for y-axis in fluorescence spectrum was incorrect,which should be "FL" but not "FI".

The label for x-axis in figure 11b was incorrect.

Reviewer 2 Report

In this manuscript, a DNA-Templated Silver Nanoclusters (DNA-Ag NCs) was developed where Cu2+ could be used to qenched the fluorescence of DNA-Ag NCs, which further achieved the turn on detection to glyphosate. This probe had "turn-on" fluorescent sensing of glyphosate and had the advantages of high sensitivity, simple operation and low cost. The work is something novelty, however, it was not well organized and there are still some issues needs to be addressed. Therefore, I suggest “Minor revision”.

1.        The “Cu2+” should be “Cu2+” in the Abstract;

2.        The binding constant experiments about (glyphosate and Cu2+) and (DNA-Ag NCs and Cu2+) should be added;

3.        Please explained the reason of the emission spectrum higher than the excitation spectrum in the Figure 6;

4.        Figure 9 B need be added to the appropriate location in this manuscript;

5.        Please explained the reason of the highest fluorescence intensity of DNA-Ag NCs in weak alkaline environment.

6.    Some Refs. may be added, such as  10.1002/anie.202013302.

Round 2

Reviewer 1 Report

I suggest that this manuscript should be accepted.

Reviewer 3 Report

The authors have addressed all my queries. The editor may accept the manuscript for publication.